# Comfort During Motion: Analyzing the Pressure Profile of Auxetic Bra Pads

**DOI:** 10.3390/ma18225071

**Published:** 2025-11-07

**Authors:** Yin-ching Keung, Kit-lun Yick, Joanne Yip, Annie Yu

**Affiliations:** School of Fashion and Textiles, The Hong Kong Polytechnic University, Hong Kong, China; doris-yinching.keung@connect.polyu.hk (Y.-c.K.); joanne.yip@polyu.edu.hk (J.Y.); annie.tw.yu@polyu.edu.hk (A.Y.)

**Keywords:** auxetic materials, sports bra pads, pressure distribution, breast support

## Abstract

Auxetic structures, characterized by a negative Poisson’s ratio and unique form-fitting deformation, are adopted for designing a bra pad that would facilitate bras with a flexible and adaptive fit. This study compares the pressure distribution between auxetic and traditional molded bra pads, highlighting the advantages of auxetic materials in applying uniform pressure and addressing health concerns. Seven athletic female participants with a bra size of 75B comprise the study sample. Anthropometric data of naked breasts are collected by using three-dimensional (3D) scanning to obtain the underbust and full bust dimensions in the standing and leaning forward positions, while the pressure distribution is measured with the Novel Pliance^®^ pressure measurement system in three poses: standing, static cycling, and dynamic cycling. The results show that the auxetic designs of bra pads consistently apply a more uniform pressure distribution compared to traditional foam pads, with mean pressures of 2.92 kPa for auxetic pads compared to 4.81 kPa for traditional foam pads during static cycling. Moreover, auxetic pads reduced maximum pressure by 25% compared to molded cups, and spatial variability was halved (SD 0.85 kPa vs. 1.70 kPa). Notably, at the bra neckline, auxetic pads exhibit increased pressure as the body leans forward, demonstrating their ability to adapt to changing breast shapes while maintaining adequate bra-breast contact. In contrast, in the lower breast area, the auxetic pads show a decrease in pressure, which indicates their capacity to accommodate variations in breast girth or volume without exerting excessive force. These findings highlight the superior adaptability and wear comfort provided by an auxetic structure, which shows its potential to address the dynamic support needs of active women. Overall, the auxetic designs of a bra pad in this study represent a significant advancement in sports bra technology and offer a promising alternative to traditional molded cups in activewear design.

## 1. Introduction

Wearing bras, particularly sports bras, provides critical support that mitigates the effects of gravity on the breast tissues, thereby reducing exercise-induced breast pain and movement, which are factors that contribute to potential tissue damage during physical activities [1]. By offering such essential support, sports bras enhance the overall wear comfort and improve the athletic performance of women who are participating in various sports activities. Bra support is provided through the exertion of a certain amount of pressure onto the breast tissues. A number of studies have explored the importance of bra support and pressure in breast biomechanics. Gilmer et al. [1] and Illidi and Jensen [2] stated that effective support and appropriate pressure from sports bras reduce exercise-induced pain and improve wear comfort and performance during sports activities. Pressure comfort is also a key factor that influences the overall satisfaction with sports bras [3]. Both insufficient and excessive pressure can cause breast discomfort, pain, and potential long-term damage [4,5,6,7]. Achieving the right balance of support and pressure is crucial for optimal breast health and wear comfort during physical activity.

Given the direct contact of bras with the body and the sensitivity of the skin to pressure and touch, the impact of pressure on bra comfort has prompted numerous studies that investigate the pressure exerted on the body by various bra components [5,8]. Most research work on bra pressure or comfort focuses on the shoulder straps or underband [3,9,10], with few that examine the pressure dynamics of the bra pads, which is a research gap in current understanding of how internal pressure from bra pads affects wear comfort and support. The distribution of the internal pressure from bra pads is closely related to fit, which is crucial for ensuring comfort during wear. However, evaluating cup fit is challenging, as the internal structure of the cup is not visible and cannot be easily determined through traditional fitting methods.

Breast tissues are soft and viscoelastic, so their response to pressure varies with breast shape. Even with a well-fitting bra in a static position, variations in breast shape and asymmetry can lead to uneven pressure distribution, thus potentially affecting wear comfort during movement [11,12,13,14]. Women with the same bra size and standardized body dimensions may still experience varying compression effects due to their unique breast shape [15]. Furthermore, pressure distribution can change with movement, as each sport involves distinct motion patterns and breast dynamics [16]. Dynamic activities or sports that involve forward-leaning, such as cycling, can lead to discomfort and exert uneven pressure. These challenges highlight the limitations of current bra designs and more accurate methods for evaluating bra-breast pressure [15]. Research needs to explore how specific bra pad features influence control over movement with different activities, as the effects can vary due to the unique dynamics of each type of sport.

Previous studies have attempted to measure and simulate bra pressure under various conditions. For instance, Lu et al. [11] measured bra pressure at five points, including the left nipple and four surrounding areas of the breast, during standing and vertical jumping across three bra conditions, alongside breast displacement data to evaluate the effectiveness of different bra materials. Sun et al. [17] simulated the interaction between breasts and various bra cup materials by using a biomechanical model based on the finite element method, which validated the contact pressure at the bra strap and bottom of the bra through measurements with the Novel Pliance-X System. Musilova et al. [15] evaluated testing methods for the pressure distribution of bra cups made of elastic polyurethane foam on soft breast tissues by directly measuring pressure at 12 points around the breast with sensors and evaluating body temperature changes through a thermography system. While these studies provide valuable insights, the variations in testing methods and limited focus on internal bra pad pressure suggest that further research is needed to fully understand this aspect of bra design.

Auxetic materials, characterized by a negative Poisson’s ratio, expand laterally when stretched, thus offering unique mechanical properties such as enhanced flexibility, energy absorption, and conformability to accommodate all body shapes [18,19]. These attributes make auxetic structures particularly suitable for sports bra pads, as they can adapt to dynamic body movements and postures, such as leaning or bending, while the bra pad maintains a uniform pressure distribution. Additionally, auxetic bra pads are shown to have the ability to accommodate various breast shapes and contours effectively, thus improving fit and stability during dynamic movement. The unique form-fitting deformation of auxetic structures allows for a flexible fit on curved surfaces [20]. However, the impact of auxetic structures and materials on the distribution of pressure from sports bra pads on the breast has yet to be fully explored.

Given the importance of pressure of sports bra cups on wear comfort and support [8,21], this study focuses on the innovative alternative to conventional molded pads by introducing additively manufactured auxetic pads whose negative-Poisson’s-ratio lattice expands laterally under load, offering superior conformability. An associated patent entitled “Adaptive Auxetic Bra Pad Design” has been filed (No. 202521251365.1, China National Intellectual Property Administration, 17 June 2025). It is hypothesized that, relative to standard molded pads, auxetic pads will (i) generate significantly lower peak contact pressure and (ii) produce a more uniform spatial distribution (lower coefficient of variation) across the breast surface under both upright standing and forward-leaning cycling postures, with larger unit-cell sizes further improving pressure homogeneity during dynamic motion. The overarching aim is, therefore, to engineer auxetic pad architectures that adapt flexibly to individual breast morphology, limit localized high-pressure zones, maintain even interfacial loading, and thereby mitigate the discomfort and potential tissue damage associated with existing sports-bra designs.

## 2. Materials and Methods

### 2.1. Participants

This investigation was designed as a controlled pilot study. Seven healthy, athletic female participants who are 25 to 35 years old were recruited for the bra pressure test. None of the participants has a history of pregnancy or breastfeeding. To maintain consistency within the sample group, all of the participants wore a bra with a band size of 34 and a full B cup size. Ethical approval was obtained prior to the commencement of the study (Approval No. HSEARS20220825001), and written consent was secured from all the participants.

Only the left breast was instrumented to permit comparison with previous unilateral breast-pressure protocols and to minimize sensor-related motion artefacts during cycling. Consequently, potential left–right differences in volume, shape or tissue stiffness were not quantified; future full-scale studies should employ bilateral measurements to verify generalizability.

### 2.2. Bra Samples

Five bra samples were used to evaluate their exerted pressure. Sample A is a commercially available cycling sports bra composed of 75% nylon and 25% spandex, and features an embedded polyurethane (PU) foam cup of uniform thickness. Samples B, C, D, and E incorporate auxetic bra pads with a uniform thickness of 3 mm, but vary in unit cell size of 6.14 mm, 7.14 mm, 8.14 mm, and 9.14 mm, respectively. The auxetic pads are fabricated from silicone elastomer (KE-1300T, Shin-Etsu Chemical Co., Ltd., Tokyo, Japan), which cures with the addition of 10% curing agent. A re-entrant auxetic structure with a bow-tie configuration was designed to model the mechanical properties of the bra pads. Prior to fabrication, CAD models of the bow-tie honeycomb structure were created with the specified unit cell dimensions, maintaining a constant line width of 1.2 mm and a void height of 0.4 mm for all unit cells. These models were developed using Autodesk Fusion 360, converted into STL files, and printed via stereolithography to produce high-resolution molds. To eliminate entrapped air, a vacuum casting technique was employed, and the specimens were cured at 23 ± 2 °C for a minimum of 12 h to ensure complete cross-linking before demolding. Detailed mechanical characterization of the lattice geometry is provided in Keung et al. (2024) [20]. The silicone component is made from silicone elastomer (KE-1300T), which cures with the addition of a 10% curing agent and is directly bonded to a 4-way stretch, tight-fitting shapewear bra top sourced from the market. Figure 1 provides a summary of the bra samples used in this study.

### 2.3. Experimental Procedure

Breast 3D scanning was performed with the Artec Leo 3D scanner, which employs laser-free structured light scanning technology (Artec Leo; Artec 3D, Luxembourg; 3D resolution, ≤0.2; 3D point accuracy, ≤0.1 mm). The Artec Leo produces highly detailed and accurate 3D images in real time, ensuring precise measurements. It has been used and validated in previous clinical work for 3D imaging [22,23]. After the scanning process, the resulting scans are imported into Artec Studio (2024, Artec 3D, Luxembourg), where the data are converted from a point cloud format into a working 3D object. For consistency in measurements, this 3D object is then exported as an OBJ file. To measure specific dimensions of underbust and full bust girth of the scanned body, OBJ files were imported into Geomagic Wrap (v2024.0, 3D systems, Inc., Rock Hill, SC, USA), analyzed for geometric properties using the built-in measurement tools and proceeded with surface-based measurement. This structured approach guarantees that the anthropometric data collected are reliable and suitable for further analysis.

Participants were scanned in a braless condition while maintaining normal breathing. Two standardized postures were recorded: (i) upright standing, with feet positioned 10 cm apart and arms relaxed at a 10° abduction, and (ii) forward leaning until the torso was horizontal, corresponding to a 90° flexion. Each posture was captured three times to ensure repeatability and accuracy. The full bust and underbust girths, which were kept level across the subject’s back, were measured for each scan. Measurements from these three replicates were averaged prior to further analysis. Figure 2 illustrates the specific locations measured for the underbust and full bust girths.

Cup size was determined by subtracting the underbust girth from the full bust girth, with a difference of 10.0 cm indicating an A cup, 12.5 cm indicating a B cup, and 15.0 cm indicating a C cup [24]. Given that breast tissue shapes can change with different body postures, cup size was calculated using measurements from both the standing and forward-leaning positions. The overlapping measurements between these two postures highlight the differences that arise from changes in position, effectively accounting for posture-related variations. A paired *t*-test analysis was conducted to compare the measurements of the full bust and underbust girths in both the standing and leaning positions.

The Novel Pliance-X^®^ pressure measurement system (Novel Electronics, Munich, Germany), which is a validated tool for measuring the pressure interface between garments and soft tissues [25], was then utilized to measure the bra-breast pressure by placing the sensors between the breast skin and the bra pads. This is a capacitive sensor with a sensing pressure range of 0.5 to 60 kPa. For improved visibility of the pressure values, the pressure range was adjusted to 0 to 10 kPa. Six strip sensors, 10 mm in diameter (in contact area of 78.54 mm^2^) and 0.95 mm in thickness, were strategically positioned and attached to the left breast of each participant. One sensor was placed at the nipple, while five additional sensors were positioned above and below the nipple, maintaining a distance of 5 cm from it, as illustrated in Figure 3. The sensors are designed for high accuracy, with a typical performance within ±1% of full-scale accuracy, thus ensuring reliable pressure readings. The participants were instructed to wear the five different bra samples under three specific conditions: (i) normal breathing for 30 s in a static standing position, (ii) normal breathing for 30 s in a cycling posture on a stationary bike with hands positioned on the drops and arms bent, and (iii) dynamic cycling in the dropped position for one minute at a controlled energy output of 25 watts, with a cadence of approximately 70 RPM ± 5 (see Figure 4). To minimize fatigue or adaptation effects, the bra pads were tested in a random order for each participant. This randomization helps ensure that the results are not biased by the order of testing, allowing for a more accurate evaluation of each pad’s performance. Throughout the cycling process, the shoulder and elbow joint angles of each athlete were carefully controlled to ensure consistency in posture. Participants were allowed to rest between conditions to ensure that bra pressure measurements were not influenced by fatigue. Calibration was performed between each condition of each sample to maintain consistency and reliability across trials. Changes in bra pressure on the breast were systematically measured under these conditions to evaluate the effects of different bra pads in the different body positions. Each of the participants completed the activities three times to ensure the repeatability and reliability of the pressure readings obtained.

Descriptive statistics were employed to measure the mean and standard deviation of the pressure evaluations for the bra pads under the different conditions. The mean and standard deviations were calculated to evaluate the pressure distribution across the six measured points, which represent different areas of the bra pad. A two-way analysis of variance (ANOVA) was conducted to evaluate the effects of the bra samples and body positions on the pressure distribution at these points. All data were analyzed by using SPSS software (IBM SPSS Statistics for Windows, version 20; IBM Corp., Armonk, NY, USA), with results deemed significant at an alpha level of *p* < 0.05.

## 3. Results

### 3.1. Anthropometric Breast Measurements Across Postures

Table 1 presents the anthropometric data collected from the seven subjects, including their underbust and full bust girth values measured in two positions: standing and leaning forward at a 90-degree angle. In the standing position, the underbust girth ranges from 73 cm to 77 cm, and full bust girth ranges from 87 cm to 90 cm, which shows a difference of 13 cm to 15 cm, with all of the subjects classified as a B cup size. In the leaning position, the underbust girth ranges from 77 cm to 88.5 cm, while the full bust girth ranges from 87.5 cm to 94 cm, which shows a difference of 3 cm to 16 cm, with cup sizes that vary from AAA to C. There are significant differences in both the underbust and full girth measurements between the standing and leaning positions, with *p*-values of 0.003 for the underbust girth and 0.001 for the full bust girth (both *p* < 0.01). These results highlight the substantial impact of body posture on the breast dimensions, which suggests that bra fit and wear comfort can vary considerably based on whether the wearer is standing or leaning. This underscores the need for bra designs that accommodate these positional changes to enhance wearability and effectiveness.

The conventional method for determining bra cup size involves calculating the difference between the underbust girth and full bust girths, a practice widely acknowledged in the commercial lingerie industry. Previous studies indicate that the contribution of the chest wall compartments to tidal volume varies with posture, thus subsequently influencing girth measurements around the chest [26]. The findings of this study reveal significant variations in the underbust and full bust measurements across the different body positions, which raises important concerns regarding current bra fitting protocols. Typically, bra fittings are conducted primarily in a standing position, which may not adequately account for the dynamic changes in breast and ribcage dimensions that occur during various activities. This issue is particularly relevant in the context of sports and physical activities, where different postures can exacerbate discrepancies in fit and support. As a result, traditional fitting methods may not yield optimal sizing and wear comfort for individuals engaged in such activities.

Table 2 lists the percentage changes in girth measurements between the standing and leaning positions and reveals notable differences in the underbust and full bust measurements across the subjects. The percentage changes for the underbust girth range from 3.4% to 14.9%, while for the full bust girth, they range from 1.7% to 5.7%. Individual variations in anatomical responses to posture likely contributed to these differences.

A wider range of percentage changes was observed in the underbust area, thus indicating that this region is more affected by the torso compression or relaxation during leaning. Beyond the substantial changes in bra cup size (Table 2), these findings underscore the need for flexible materials in the underband of bras, which would allow for rib cage expansion and ensure the band remains secure with various postures. The variability in underbust measurements highlights the different requirements for stretchability in bra design. In contrast, the smaller percentage changes in full bust measurements reflect how breast tissues respond differently to positional shifts. While the breasts are supported by the pectoral muscles and rib cage, they may be less directly influenced by rib cage expansion during breathing. Instead, changing postures primarily affects breast tissue through alterations in morphology and redistribution. Bra design, therefore, must accommodate both the static measurements and dynamic behavior of breast tissues.

### 3.2. Pressure Distribution of Bras Across Six Points

Table 1 presents the anthropometric data collected from the seven subjects, including the underbust and full bust girth values measured in two conditions: standing and leaning forward at a 90-degree angle. In the standing position, the underbust girth ranges from 73 cm to 77 cm, and full bust girth ranges from 87 cm to 90 cm, which are differences of 13 cm to 15 cm, with all subjects classified as a B cup size. In the leaning position, the underbust girth ranges from 77 cm to 88.5 cm, while the full bust girth ranges from 87.5 cm to 94 cm, thus resulting in a difference of 3 cm to 16 cm, with cup sizes varying from AAA to C.

The pressure distribution analysis of the five bra samples under the various conditions showed significant differences between the molded pad (Bra A) and auxetic pads (Bras B–E). The radar charts revealed that Bras B–E have similar, more uniform pressure patterns across the six points, while Bra A shows a distinctly different and inconsistent distribution among the subjects, which indicates a less reliable fit and support. Most of the subjects who wore Bras B–E experienced consistent pressure, except for Subject 2, who provided notably different readings for Point 4. These findings suggest that auxetic bra pad structures generally offer more consistent wear comfort and support, although individual anatomical differences can still affect the fit and pressure distribution.

#### 3.2.1. Standing Pose

Figure 5a shows that Bra A has significantly higher and more varying pressure values across the key measured points. For instance, Subject 4 recorded 3.7 kPa at Point 5, and Subjects 2 and 6 have elevated pressures of 2.7 kPa at Points 4 and 6, respectively, thus indicating concentrated areas of pressure and inconsistent support. In contrast, Bra B exhibits lower and more uniform pressure distribution, with only minor exceptions of Subject 2, who indicated higher pressure at Point 1 and Subject 4 at Point 4, but overall, they showed similar patterns across all points of pressure. Bras C, D, and E exhibit radar shapes similar to that of Bra B; thus, they exert relatively less pressure. The heatmap presented as Figure 6 further corroborates these findings by depicting Bra A in red and orange shades, which signify regions of extreme pressure. Conversely, Bras B to E display increasing areas of blue, indicating a more uniform pressure distribution and lower pressure levels. The overall color intensity lightens from Bra B to Bra E, correlating with the transition from smaller to larger auxetic cell sizes. These results suggest that the auxetic bra pads (Bras B to E) facilitate consistent support and enhanced wear comfort, whereas the molded pad (Bra A) is associated with localized pressure, resulting in a less uniform fit.

#### 3.2.2. Static Cycling Pose

In Figure 5b, Bra A continues to show elevated and varying pressure values during static cycling, with Subject 4 recording high pressure at Point 5, Subjects 3 and 7 at Point 4, and Subjects 2 and 6 at Point 6. This shift in pressure distribution compared to standing suggests that the support and fit of Bra A cannot accommodate the changes in body posture. In contrast, Bras B to E maintain similar, uniform pressure patterns as those of the standing position, thus indicating stable and effective support across all six points. Figure 6 reinforces this observation, presenting a heatmap that displays increasing blue shades from Bras B to E, which signifies their ability to consistently manage pressure, even as body posture changes. The colors become more uniform as the auxetic cell size increases from Bra B to Bra E, further illustrating the enhanced pressure distribution provided by the larger cell dimensions.

#### 3.2.3. Dynamic Cycling Pose

Figure 5c shows that during dynamic cycling, Bra A produces varying pressure results, with Subjects 2, 3, and 7 experiencing high pressure at Point 6, and Subjects 1 and 4 at Point 5. This highlights a non-uniform pressure distribution across the subjects and poses, as illustrated in Figure 6, where Bra A is depicted with a mix of red and orange areas, indicating localized pressure points across subjects and poses. In contrast, Bras B to E maintain consistent and uniform pressure patterns, even though Bras C to E have slightly higher overall values. The heatmap shows these bras predominantly in blue, indicating effective pressure management. This consistency indicates that Bras B to E effectively offer both support and adaptability, which means they respond well to body movement during cycling. Their design ensures optimal pressure management, thus providing extensive bra-breast contact and adaptive support regardless of body orientation and activity.

### 3.3. Comparative Pressure Performance of Bras Under Various Poses

Table 3 summarizes the mean and maximum pressure values, along with standard deviations and 95% Confidence Intervals (CIs), for Bras A to E during standing, static cycling, and dynamic cycling. All of the bras show moderate mean pressures (1–2 kPa), but Bra A consistently records the highest maximum pressure: 3.69 kPa during standing, 4.81 kPa during static cycling, and 4.03 kPa during dynamic cycling. This indicates a non-uniform pressure distribution, with values that are highly concentrated in certain areas. The maximum pressure values are the lowest during standing and increase with activity and peaking during dynamic cycling. Bra A shows the highest pressure during static cycling, while Bras B to E show a steady rise from standing to dynamic cycling, with the pressure during dynamic cycling exceeding 3 kPa.

An examination of the auxetic pads reveals that increasing the cell size from 6.14 mm in Bra B to 9.14 mm in Bra E is associated with a slight decrease in maximum pressure across most conditions; however, this trend is not evident during dynamic cycling. Bra A, a commercial molded bra, records the highest maximum pressures: 3.69 kPa during standing and peaking at 4.81 kPa during static cycling. In contrast, the auxetic bras (B to E) generally exhibit maximum pressures below 3.1 kPa during standing and static cycling conditions. Notably, during dynamic cycling, maximum pressures remain similar across the different bras, suggesting that cell size may have a limited impact on pressure distribution in this context. While larger cell dimensions in the auxetic pads may enhance wear comfort by reducing localized pressure in certain scenarios, their overall effect on maximum pressure during dynamic activities appears to be less pronounced. Nevertheless, the auxetic behavior, influenced by cell geometry, still contributes to improved force distribution, making these materials particularly beneficial for bras that require uniform support [27,28].

Standard deviations further highlight the pressure variability. Bra A has significantly higher standard deviations (1.23, 1.70, and 1.40 for standing, static cycling, and dynamic cycling, respectively) compared to Bras B–E. These differences also indicate non-uniform pressure distribution, which can cause discomfort and localized pressure points during prolonged wear. In contrast, Bras B to E exert constant pressure, thus supporting their effectiveness in providing consistent wear comfort and support with different activities and poses.

The results indicate significant variations in maximum pressure values among the different bra samples (A to E) across various conditions: standing, static cycling, and dynamic cycling. Bra A consistently recorded the highest maximum pressures, peaking at 4.81 kPa during static cycling, while also exhibiting considerable variability, as reflected in its 95% Confidence Intervals (CIs). In contrast, Bras B to E demonstrated more consistent pressure distributions, with mean maximum pressures remaining below 3.1 kPa across all conditions. Notably, the CIs for Bra A during static cycling indicate a wide range of potential values, suggesting that while it may provide substantial support, it could also lead to discomfort due to localized pressure points. Conversely, Bras B to E exhibited narrower CIs, indicating more uniform pressure distribution and potentially greater comfort during prolonged wear.

To examine how the body posture affects pressure distribution, Figure 7 compares the maximum pressure values of the six measured points for Bras A (molded pad) and D (with auxetic pad, 8.14 mm cell size) based on Subject 4 as a representative case. The two pads show distinct pressure responses when Subject 4 transitioned from standing to leaning forward during static cycling.

Bra A exhibits exceptionally high pressure at Point 5, which increases from 3.69 kPa (standing) to 4.81 kPa (static cycling). On the other hand, the pressure at Points 1, 2, and 3 decreases when transitioning from standing to cycling. Points 4, 5, and 6 show variable increases. This pattern reflects the fixed shape of the molded pad, which cannot readily adapt to changes in the breast morphology caused by forward-leaning and gravitational force. As the body leans forward, the breasts shift downward, often creating gaps in the upper breast area of the bra and causing the pad to lose contact with the body, especially at the neckline. This can compromise both wear comfort and aerodynamic efficiency, as a poorly fitting neckline means that the bra would not fit well to the body.

In contrast, the auxetic pad of Bra D shows more adaptive pressure. The highest pressure is at Point 1 (nipple position), which increases from 1.94 kPa (standing) to 2.29 kPa (cycling). There is also an increase at Points 2 and 3, and a slight increase at Point 4. Notably, Points 5 and 6 show decreased pressure during cycling. The auxetic structure of Bra D allows it to stretch and conform to the changing shape of the breast, thus maintaining support and close contact at the neckline as the body leans forward. This adaptability is crucial for cycling, where a secure fit at the neckline enhances aerodynamic performance.

Points 4 to 6 are in the lower breast area, with Point 5 at the midpoint where the breast volume is concentrated. For Bra A, the high pressure at Point 5 results from the breast tissues gathering in the lower area as the rib cage expands and the body bends forward. The rigidity of the molded pad of Bra A leads to significant pressure increases, which can cause discomfort with prolonged activity. In contrast, the auxetic pad of Bra D accommodates changes in girth and volume, thus maintaining support without excessive pressure.

Table 4 presents the results of a two-way ANOVA that evaluates the differences in maximum pressure among the five bra samples across the three conditions. The analysis reveals that the static cycling position has a statistically significant effect on maximum pressure values, with a *p*-value of 0.03 (*p* < 0.05). This finding indicates that the pressure exerted by the bras during static cycling is notably different, thus highlighting the influence of bra design and material on wear comfort and support during this activity.

Although variations in the maximum pressure during dynamic cycling were observed among the bra samples, these differences are not statistically significant (*p* = 0.079). This may be due to the nature of dynamic cycling, which involves continuous movement and shifting body positions, resulting in a more uniform pressure distribution across the bra and reducing the impact of specific design features. In contrast, static cycling, with a consistent posture, makes the supportive qualities of each bra more apparent. These findings highlight the importance of considering the type of activity when evaluating bra performance and wear comfort. Traditional sports bras with molded foam cups offer a structured shape and smooth silhouette, but their rigidity can limit flexibility during dynamic movements, thus potentially causing discomfort and non-uniform pressure. Auxetic bra pads, however, expand in response to pressure, which allows them to adapt to body movements, thus conforming more closely to breast shape and distributing pressure more evenly. Auxetic pads also feature breathable structures that improve airflow, so that moisture and body temperature can be better managed during intense exercise—an essential benefit for athletes. Additionally, bras with auxetic pads can be designed with adjustable straps and a wide underband to further improve stability and fit, thus ensuring the bra remains secure during a plethora of movements.

### 3.4. Performance of Pressure Distribution in Bra Pads

The analysis of the pressure distribution across the six measured points reveals significant differences in performance between the molded and auxetic bra pads, particularly at Points 2 (upper cup area) and 6 (side of the breast), with *p*-values of <0.001 and 0.006, respectively. These results highlight the importance of targeted design features in these regions to enhance wear comfort and fit. The ability of the pad to contour and flexibly adapt to the natural shape of the breasts is critical at Point 2; inadequate contouring may lead to poor support and discomfort, while effective contouring ensures a uniform pressure distribution. At Point 6, the shape and rigidity of the pad are essential for providing sufficient side support. Pads with a rigid, inflexible design may fail to accommodate natural breast deformations, especially when leaning forward, thus resulting in shifting pressure and discomfort.

The study also found that body pose significantly affects pressure distribution at Points 2 and 3, which are both located in the upper breast area. Leaning forward causes breast deformation, which can compromise the fit of the bra at the neckline, thus leading to gapping, discomfort, and inadequate coverage. This underscores the need to consider body posture when evaluating bra performance, as certain positions can exacerbate fit issues at key pressure points. In contrast, no significant differences were observed at Points 1, 4, and 5, which indicates that pressure levels in these areas remain stable across different conditions and may be less sensitive to changes in posture. Understanding the impact of body posture on pressure distribution is essential for developing bras that offer optimal fit and stability, which would ultimately improve the experience of the user during physical activities.

## 4. Discussion

This study provides critical insights into the influence of body posture on breast dimensions and pressure distribution in sports bras, thus highlighting the importance of ergonomic design in this context, laying the groundwork for future research focused on developing practical design recommendations. Our findings show that anthropometric measurements significantly vary between the standing and leaning positions, which underscores the need for bra designs that effectively accommodate dynamic changes in breast morphology. Previous research has established the significance of accurate bra fitting, particularly for athletes, as traditional fitting methods, typically conducted in static positions, might not yield optimal outcomes for active individuals [29]. The observed differences in underbust and full bust girth measurements (*p* < 0.01) corroborate existing literature, which emphasizes how variations in posture can affect breast dimensions and, consequently, bra performance [9,20].

The analysis of the pressure distribution across the different types of bras supports our hypothesis that auxetic materials enhance both wear comfort and fit when compared to traditional molded pads. The auxetic bras (Bras B to E) consistently exhibit more uniform pressure distribution and reduced localized pressure points, particularly during dynamic activities such as cycling. For instance, a previous study demonstrated that auxetic structures significantly reduce peak pressures, with instances of uniform pressure distribution improving by 30% compared to conventional designs [20]. This finding aligns with prior studies indicating that auxetic materials can distribute forces more evenly, minimizing discomfort and enhancing overall support.

The geometric configuration of the auxetic pads plays a crucial role in their mechanical behavior, as larger auxetic cell sizes facilitate greater in-plane expansion and lower stiffness, key factors in their ability to conform to body contours. This behavior is explained by the principles of material deformation, particularly the Poisson effect, where an increase in lateral dimensions occurs under axial loading. In auxetic materials, which exhibit a negative Poisson’s ratio (NPR), enhanced deformation capabilities lead to improved load redistribution and reduced pressure peaks. Previous research illustrates this relationship, showing that auxetic pads demonstrate lower PR values with increasing elongation, indicating that smaller unit cell dimensions correlate with increased NPR [20]. Notably, auxetic pads maintain their NPR characteristics even up to 100% elongation, which supports the notion that larger auxetic cells can effectively manage pressure distribution under dynamic loading conditions. Thus, integrating these material deformation principles underscores the importance of auxetic geometry in achieving optimal comfort and support in bra design.

Moreover, results indicate that the effectiveness of the auxetic pads is particularly pronounced during conditions that simulate the forward-leaning posture typical in cycling. This adaptability is essential, as inadequate breast support during physical activities can lead to discomfort, which negatively affects performance. The significant changes in the girth (3.4% to 14.9% for the underbust girth) emphasize the need for flexible materials in bra designs, especially the underband, which must accommodate rib cage expansion as the lungs expand and contract during exercise.

Additionally, the incorporation of larger auxetic cell sizes in bra pads facilitates improved load redistribution and pressure management. As noted in our previous work, larger auxetic cells allow for greater in-plane expansion and lower stiffness, enabling the material to conform more effectively to the contours of the body [20]. This mechanism is essential for ensuring that pressure peaks are minimized, thereby enhancing comfort during dynamic movements.

To take this study further, future research work should explore the long-term effects of wearing bras with auxetic materials for a range of different sports contexts, which focus on performance outcomes and user satisfaction. It is important to acknowledge several limitations in our current study. Firstly, the small sample size may limit the generalizability of our findings. Additionally, we conducted a single breast measurement per participant, which may not capture the full variability in breast morphology. The limited sensor density could also impact the accuracy of our pressure distribution measurements. Furthermore, potential sensor drift during data collection must be considered, as it may affect the reliability of the results. Lastly, the influence of factors such as sweating or temperature on pressure distribution remains untested and could impact user comfort. Investigating the impact of various auxetic structures and materials on pressure distribution could yield further insights. Additionally, expanding the participant demographics to include a wider range of body types and sizes will enhance the applicability of these findings.

## 5. Conclusions

This study highlights the crucial role of bra design in meeting the diverse needs of women, especially in terms of their breast girth and body movement. Molded bras often create fitting challenges, as women with the same bra size have different breast shapes. Additionally, changes in body girth during standing versus leaning indicate the need for bra pads that flexibly contour to these differences, thus preventing excess pressure and discomfort. This underscores the importance of adaptable designs that accommodate individual differences.

Quantitatively, auxetic pad C, with an 8.14 mm cell size, reduced maximum pressure by 25% compared to the commercial molded cup during static cycling (2.92 kPa vs. 4.81 kPa, *p* = 0.003) and halved spatial variability (SD 0.85 kPa vs. 1.70 kPa). Across all three tasks, auxetic pads kept peak pressures below 3.1 kPa, whereas the molded cup consistently exceeded 3.6 kPa, confirming that rigid foam concentrates force. Molded bras, therefore, create fitting challenges, as women with identical bra sizes exhibit different breast shapes, and standing-to-leaning girth changes of 3–15% demand pads that flexibly contour to prevent excess pressure.

The results in this study show that static cycling creates statistically significant differences in maximum pressure among the bra samples, more so than dynamic cycling. Static positions provide a clearer differentiation in support provided by the various designs. However, during dynamic cycling, constant movement and shifting body positions cause breast deformation, so that flexible and adaptive bra pads become essential for maintaining support and wear comfort. Specifically, while static cycling produced the clearest differences between samples, dynamic cycling yielded similar peak pressures across all auxetic geometries, indicating that cell-size benefits plateau under high-inertial conditions. Nevertheless, auxetic pads maintained lower mean pressures (approximately 0.2–0.4 kPa reduction) and narrower 95% confidence intervals as cell size increased, demonstrating more repeatable load sharing.

Auxetic bra pads represent an early-stage demonstration of potential benefits for sports bra comfort, pending further validation through larger studies. Unlike traditional molded cups with fixed shapes, auxetic materials expand in all directions when stretched, thus allowing the pads to conform seamlessly to the breast contours. This adaptability reduces localized pressure points and discomfort, especially during physical activities where traditional pads may shift or bunch. Auxetic pads dynamically adjust to changes in body position, thus ensuring a secure fit and adequate support during movements such as leaning or posture shifts, thereby enhancing overall wear comfort.

Overall, the research work here emphasizes the importance of the adaptability of bra pads for sports and activities that involve frequent posture changes. Auxetic pads offer unique benefits, so that they are a promising alternative to traditional molded cups and pave the way for future innovations in bra design that prioritize both functionality and wear comfort. While the sample is small (n = 7, 34B), the study provides valuable insights into pressure distribution and support during activities that involve leaning. To fully understand breast deformation across various body positions, further research with a larger participant pool is needed. Future studies should recruit a larger, size-diverse cohort, map both breasts with high-resolution arrays, and validate performance under prolonged outdoor conditions. Additionally, future work should focus on refining auxetic geometries, testing various body sizes, validating comfort perception through user surveys, and assessing durability and thermal comfort. Long-term field trials across multiple sports are necessary to relate these immediate pressure benefits to objective performance gains and subjective comfort ratings, thus contributing to the development of more effective and adaptable bra designs tailored to individual needs.

## 6. Patents

A patent titled “Adaptive Auxetic Bra Pad Design” was filed on 17 June 2025 (No. 202521251365.1) to the China National Intellectual Property Administration.

## Figures and Tables

**Figure 1 materials-18-05071-f001:**
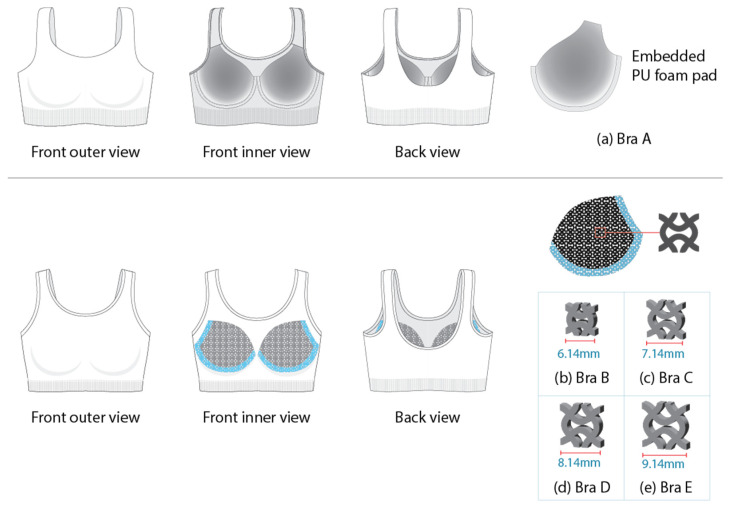
Bra samples: (**a**) Bra A is a commercial bra with a molded pad; while Bras B to E are the developed samples with auxetic pads of varying cell sizes: (**b**) 6.14 mm (Bra B), (**c**) 7.14 mm (Bra C), (**d**) 8.14 mm (Bra D), and (**e**) 9.14 mm (Bra E).

**Figure 2 materials-18-05071-f002:**
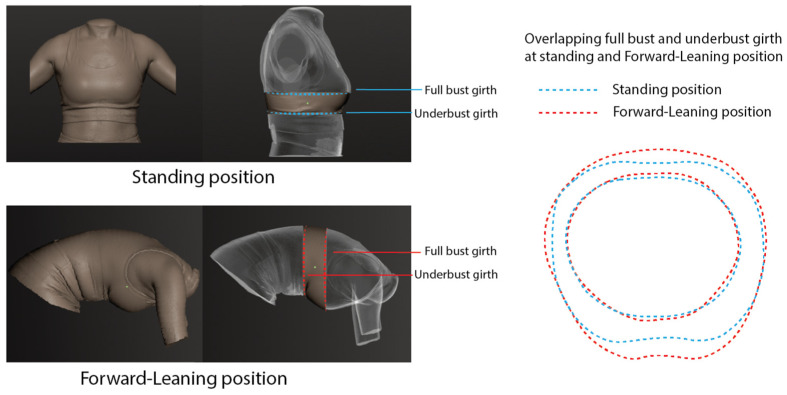
Illustration of full bust and underbust girths in standing and forward-leaning positions.

**Figure 3 materials-18-05071-f003:**
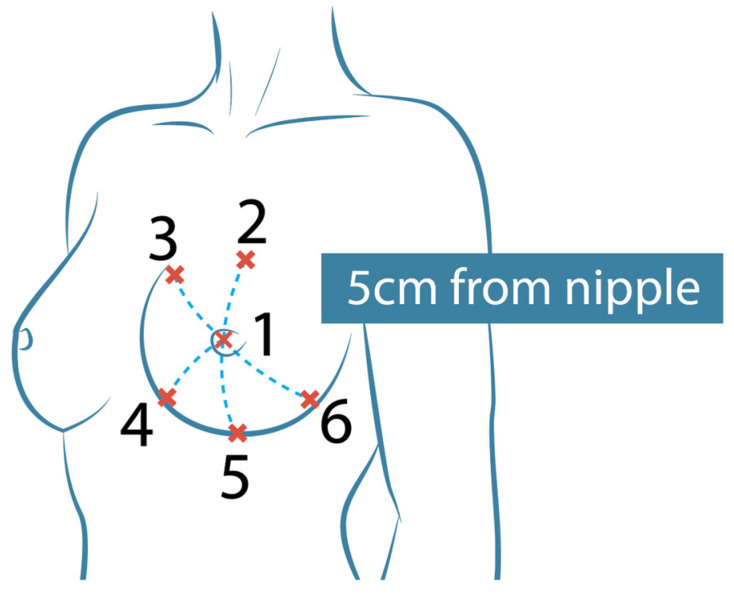
Measured pressure points.

**Figure 4 materials-18-05071-f004:**
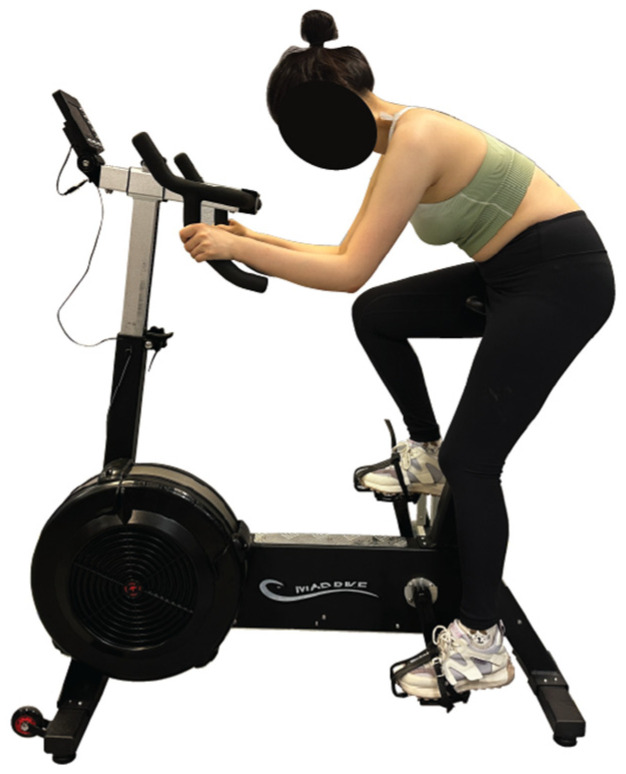
Dropped position during cycling.

**Figure 5 materials-18-05071-f005:**
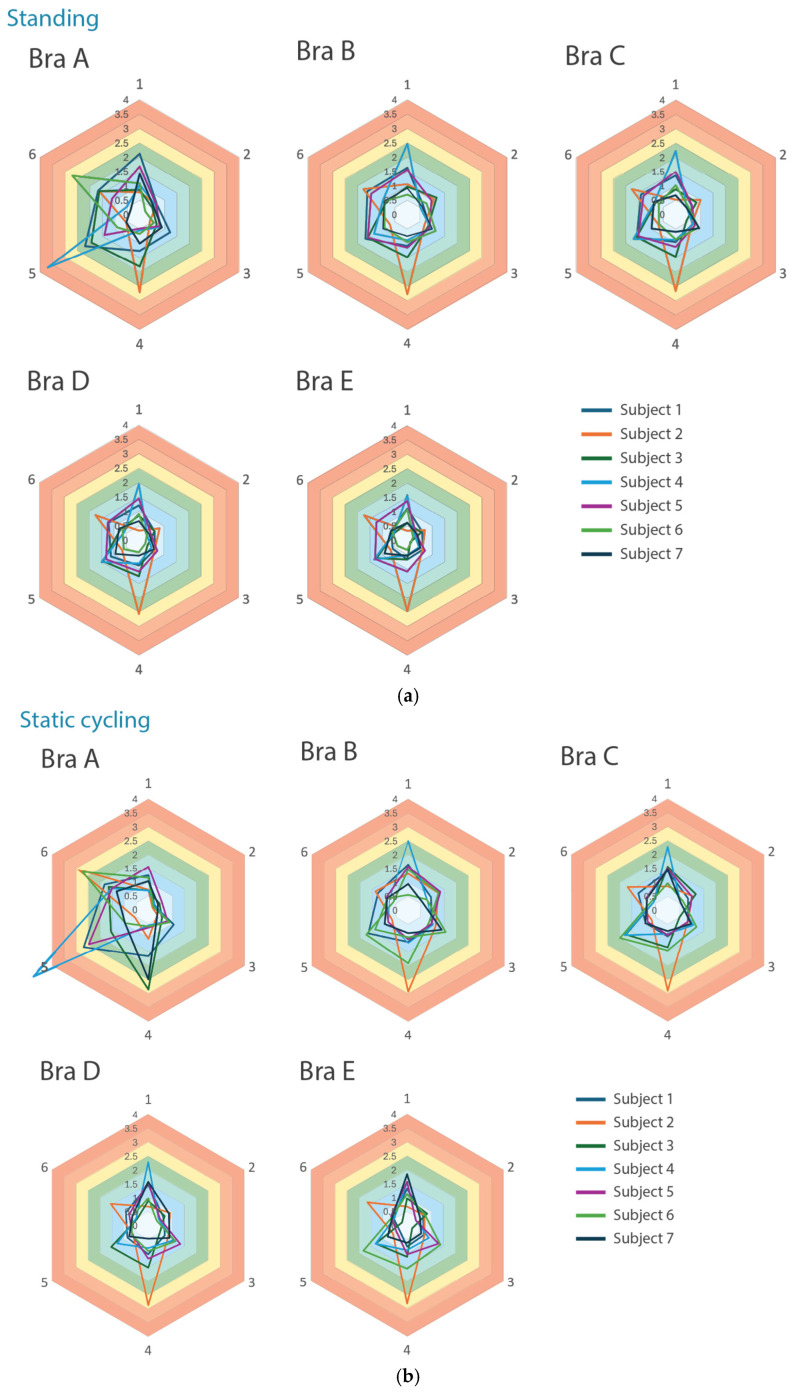
(**a**) Pressure distribution of 6 measured points during standing. (**b**) Pressure distribution of 6 measured points during static cycling. (**c**) Pressure distribution of 6 measured points during dynamic cycling.

**Figure 6 materials-18-05071-f006:**
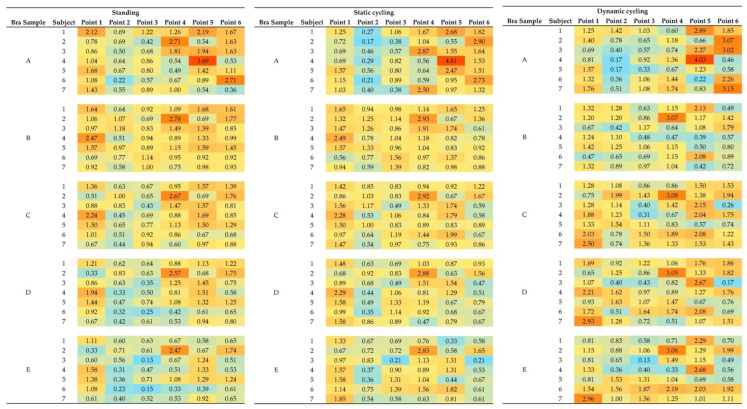
Heatmap of Pressure Distribution Across Six Measuring Points for Bra Samples A to E in Seven Subjects, with warmer colors (red) indicate high pressure values, while cooler colors (blue) indicate low pressure values.

**Figure 7 materials-18-05071-f007:**
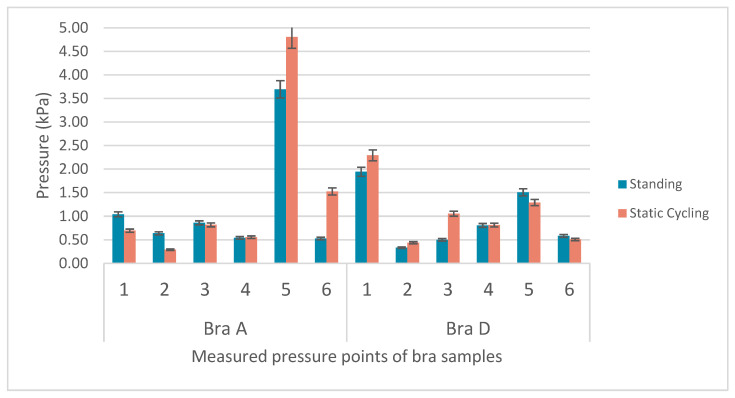
Comparison of maximum pressure values of 6 measured points for Bras A and B during standing and static cycling.

**Table 1 materials-18-05071-t001:** Breast anthropometric measurements (in cm).

Subject	Standing				Leaning			
	Underbust Girth (cm)	Full Bust Girth (cm)	Difference (cm)	Cup Size	Underbust Girth (cm)	Full Bust Girth (cm)	Difference (cm)	Cup Size
1	76.5	90	13.5	B	80	93	13	B
2	77	90	13	B	88.5	91.5	3	AAA
3	74	87	13	B	79.5	90	11	A
4	74	88	14	B	79	90	11	A
5	73	86	13	B	77.5	87.5	10	A
6	75	90	15	B	79	94	15	C
7	74.5	88	13.5	B	77	93	16	C

**Table 2 materials-18-05071-t002:** Percentage Changes in Girth Measurements Between Standing and Leaning Positions.

Subject	Under Bust			Full Bust		
	Standing (cm)	Leaning (cm)	Percentage of Change (%)	Standing (cm)	Leaning (cm)	Percentage of Change (%)
1	76.5	80	4.6%	90	93	3.3%
2	77	88.5	14.9%	90	91.5	1.7%
3	74	79.5	7.4%	87	90	3.4%
4	74	79	6.8%	88	90	2.3%
5	73	77.5	6.2%	86	87.5	1.7%
6	75	79	5.3%	90	94	4.4%
7	74.5	77	3.4%	88	93	5.7%

**Table 3 materials-18-05071-t003:** Descriptive statistics: mean, maximum pressure, standard deviation and 95% Confidence Intervals for Bras A to E under three conditions.

Condition	Bra Sample	Mean Pressure (kPa)	Maximum Pressure (kPa)	Standard Deviation	95% CI (kPa)
Standing	A	1.53	3.69	1.23	0.972, 2.088)
	B	1.34	2.78	0.805	(0.973, 1.707)
	C	1.21	2.67	0.841	(0.826, 1.594)
	D	1.13	2.57	0.852	(0.742, 1.518)
	E	1.09	2.47	0.832	(0.713, 1.467)
Static cycling	A	1.46	4.81	1.70	(0.688, 2.232)
	B	1.45	2.93	0.769	(1.100, 1.800)
	C	1.33	2.92	0.851	(0.946, 1.714)
	D	1.25	2.88	0.864	(0.858, 1.642)
	E	1.21	2.83	0.893	(0.817, 1.603)
Dynamic cycling	A	1.51	4.03	1.40	(0.875, 2.145)
	B	1.49	3.07	0.794	(1.130, 1.850)
	C	1.76	3.08	0.789	(1.403, 2.117)
	D	1.49	3.05	0.912	(1.077, 1.903)
	E	1.85	3.06	0.930	(1.435, 2.265)

**Table 4 materials-18-05071-t004:** ANOVA—Maximum_pressure.

		Sum of Squares	df	Mean Square	F	*p*
Bra_sample	Standing	3.78	4	0.945	2.34	0.078
	Static cycling	7.59	4	1.896	4.97	0.003
	Dynamic cycling	3.71	4	0.928	2.33	0.079
Residuals	Standing	12.10	30	0.403		
	Static cycling	11.45	30	0.382		
	Dynamic cycling	11.97	30	0.399		

## Data Availability

The original contributions presented in this study are included in the article. Further inquiries can be directed to the corresponding author.

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
