# Peer review of "Comfort During Motion: Analyzing the Pressure Profile of Auxetic Bra Pads"

_materials, 2025, doi:10.3390/ma18225071_

Round 1
Reviewer 1 Report
Comments and Suggestions for Authors
Dear Authors,
The topic of this article is very interesting, but above all, utilitarian. In my opinion, the methodology and some of the discussion need improvement. This will certainly enhance the scientific value of this article.
Detailed comments below:
Line 124: Each piece of equipment used in the research should be described (model: manufacturer, city, country). Please review the entire research methodology from this perspective and fill in any gaps.
Line 135: There is no reference to this method. It should be thoroughly explained or add a footnote.
Line 161: The software must also be described in detail, just like the equipment itself.
In my opinion, the methodology should be described in more detail. The research should be reproducible. This is undoubtedly a weakness of this work.
Line 195: Table 5: I must admit that an accuracy of up to 0.5 cm, in scientific research, when the obtained values ​​are at the level of several or even several dozen cm, is very low. Furthermore, I don't see average values. How many measurement repetitions were performed? Furthermore, how were the measurements performed? Were these data from 3D images taken with a scanner? Were they quality-verified?
Line 267, figure 5 c. In my opinion, the use of these types of graphs (radar graphs) poorly illustrates the presented results. These types of graphs are often used to analyze data collected from subjective evaluations, e.g., food testing by a group of testers (this is just an example). Differences are difficult to see, and the graphs are difficult to read. I think the bar graphs were better. However, I don't suggest any changes; perhaps these could be further refined.
Line 279: The values ​​are already in mm. You should standardize the units.
Figure 6: You should use error bars on the graph (if possible).
Line 373: The discussion is very short and lacks any references to the literature. This is essentially just a summary.
Line 406: The conclusions encompass the entire work and appear well-formulated.
The literature is sparse. You should cite more publications.
Reviewer 2 Report
Comments and Suggestions for Authors
The manuscript is well-structured, the research question is timely and relevant, and the experimental design is thoughtfully conceived. The use of auxetic materials in bra pad design is a novel approach with clear potential to advance both academic understanding and practical applications in sportswear and intimate apparel. The clarity of writing and the logical flow of the manuscript make it accessible to a broad scientific audience.
The visual presentation of the manuscript is, for the most part, highly professional. All figures (except for Figure 6) are clearly prepared, well-labeled, and consistent with the standards expected in scientific publications. However, Figure 6 stands out as an exception: it is obviously a screenshot, which detracts from the overall professionalism of the paper. Recreate Figure 6 using professional graph-creating software.
As a minor suggestion, the authors could consider explicitly stating the research hypothesis in the introduction.
The comprehensive pressure distribution and anthropometric measurements presented in this study provide valuable quantitative data on how both auxetic and traditional bra pads interact with the breast under various postures and activities. I recommend that the authors highlight in their discussion or future work section that these experimentally derived values (such as the mean and maximum pressure at multiple anatomical points, as well as breast girth changes across postures) could serve as realistic boundary conditions and validation benchmarks for future finite element analysis (FEA) studies. Incorporating these empirical datasets into FEA simulations would enable more accurate biomechanical modeling of bra fit and breast deformation, thereby advancing both computational and experimental research in sports bra design.
Reviewer 3 Report
Comments and Suggestions for Authors
Dear Author,
Please find attached a document with my comments on the manuscript. I hope these observations are helpful in further improving the work.
Best regards

Reviewer 4 Report
Comments and Suggestions for Authors
Here are some comments about your work:
-In the abstract, it is recommended to include numerical values (e.g., mean and maximum pressures, percentage reductions) to provide a clearer quantitative description of the results.
-The authors should clearly state what is fundamentally new in this study compared to their previous work on auxetic bra pad design.
-The study includes only seven participants, all with the same bra size (34B), which is insufficient for reliable statistical inference given the anatomical variability of breast shape. Please justify this limitation and discuss its impact on the validity of the conclusions.
-The text repeats descriptive findings without offering a mechanical or structural analysis of the auxetic behavior (e.g., lateral expansion, load redistribution, or the effect of a negative Poisson’s ratio). In my opinion, a modeling or simulation component—or at least a conceptual framework that links unit-cell geometry, deformation, and pressure distribution—is needed to enhance the scientific depth of the study.
-The manuscript lacks material characterization (mechanical properties, Poisson’s ratio, modulus, or density) necessary to support a study claiming to address a mechanical metamaterial topic such as auxetic geometries.
-Since silicone pads exhibit strain rate–dependent behavior, a viscoelastic characterization (e.g., dynamic mechanical analysis or stress relaxation tests) is strongly recommended to explain the dynamic response observed in the pressure measurements and to correlate material properties with comfort performance.
The manuscript would benefit from a thorough English language revision to improve clarity, grammar, and conciseness.
Round 2
Reviewer 1 Report
Comments and Suggestions for Authors
Dear Authors,
I accept all changes. I believe this article is now ready for publication.
Reviewer 3 Report
Comments and Suggestions for Authors
Dear Author,
Thank you for addressing the comments provided. I wish you success with this and your future projects.
Best regards
Reviewer 4 Report
Comments and Suggestions for Authors
The authors have corrected the manuscript.